# RORα Suppresses Cancer-Associated Inflammation by Repressing Respiratory Complex I-Dependent ROS Generation

**DOI:** 10.3390/ijms221910665

**Published:** 2021-10-01

**Authors:** Wei Mao, Gaofeng Xiong, Yuanyuan Wu, Chi Wang, Daret St. Clair, Jia-Da Li, Ren Xu

**Affiliations:** 1Markey Cancer Center, University of Kentucky, Lexington, KY 40536, USA; Wei.Mao@uky.edu (W.M.); gaofeng.xiong@uky.edu (G.X.); ywu244@uky.edu (Y.W.); chi.wang@uky.edu (C.W.); daret.stclair@uky.edu (D.S.C.); 2Hunan International Scientific and Technological Cooperation Base of Animal Models for Human Disease, School of Life Sciences, Central South University, Changsha 410078, China; lijiada@sklmg.edu.cn; 3Department of Pharmacology and Nutritional Sciences, University of Kentucky, Lexington, KY 40536, USA; 4Department of Molecular and Cellular Biochemistry, University of Kentucky, Lexington, KY 40536, USA; 5Department of Toxicology and Cancer Biology, College of Medicine, University of Kentucky, Lexington, KY 40536, USA

**Keywords:** tumor microenvironment, reactive oxygen species, macrophage, breast cancer, orphan nuclear receptor, complex I

## Abstract

Breast cancer development is associated with macrophage infiltration and differentiation in the tumor microenvironment. Our previous study highlights the crucial function of reactive oxygen species (ROS) in enhancing macrophage infiltration during the disruption of mammary tissue polarity. However, the regulation of ROS and ROS-associated macrophage infiltration in breast cancer has not been fully determined. Previous studies identified retinoid orphan nuclear receptor alpha (RORα) as a potential tumor suppressor in human breast cancer. In the present study, we showed that retinoid orphan nuclear receptor alpha (RORα) significantly decreased ROS levels and inhibited ROS-mediated cytokine expression in breast cancer cells. RORα expression in mammary epithelial cells inhibited macrophage infiltration by repressing ROS generation in the co-culture assay. Using gene co-expression and chromatin immunoprecipitation (ChIP) analyses, we identified complex I subunits NDUFS6 and NDUFA11 as RORα targets that mediated its function in suppressing superoxide generation in mitochondria. Notably, the expression of RORα in 4T1 cells significantly inhibited cancer metastasis, reduced macrophage accumulation, and enhanced M1-like macrophage differentiation in tumor tissue. In addition, reduced RORα expression in breast cancer tissue was associated with an increased incidence of cancer metastasis. These results provide additional insights into cancer-associated inflammation, and identify RORα as a potential target to suppress ROS-induced mammary tumor progression.

## 1. Introduction

Inflammation is a hallmark of cancer, featured by infiltration of tumor-associated macrophages. Macrophages are the most abundant immune-related stromal cells in the tumor microenvironment [1]. The accumulation of macrophages in tumors correlates with poor prognosis in breast cancer patients and drives cancer development and progression by inducing angiogenesis and suppressing immune response [2,3,4]. Macrophage accumulation occurs at an early stage of breast cancer development [5,6]. Therefore, the inhibition of macrophage-induced chronic inflammation may offer a promising strategy to prevent or repress cancer progression.

The disruption of the polarized epithelial tissue structure is an event that occurs at the early stage of breast cancer development [7,8], which is accompanied by a significant remodeling of the tissue microenvironment. We showed previously that the disruption of the polarized acinar structure was accompanied by increased production of (ROS) [9]. Treatment with antioxidant agents reduces ROS levels and reprograms non-polarized breast cancer cells to form polarized spheroids in a three-dimensional (3D) culture [9], indicating that the elevation of ROS is necessary to disrupt polarized acinar formation. ROS and associated oxidative stress are the driving force of cancer development and progression [10,11,12]. ROS are byproducts of normal metabolism through the electron transport chain [13]. It was shown that electron leakage from complex I and III in the mitochondria is the major source of ROS [14,15].

Increased ROS generation is associated with inflammation and macrophage infiltration [16,17]. In the 3D co-culture assay, we showed that reducing ROS levels in breast cancer cells inhibited macrophage infiltration [9]. These results suggest a potential link between the disruption of tissue polarity, ROS production, and cancer-associated inflammation. Mitochondria are considered the major source of ROS. However, how ROS production in mitochondria contributes to cancer-associated macrophage infiltration is not completely understood.

RORα is a member of the orphan nuclear receptor factor family and regulates gene expression by binding to ROR response elements (ROREs) [18]. The *RORA* gene maps to 15q22.2, a large common fragile site that is often deleted in cancer cells [19]. There are four isoforms being identified for human RORα proteins, and we showed previously that isoform 1 and 4 of RORα were expressed in human mammary epithelial cells [20]. By comparing gene expression profiles between polarized and disorganized mammary organoids, we have identified RORα as a positive regulator of tissue polarity [20]. RORα expression is reduced during cancer development [19,20,21]. Notably, the reduced RORα expression in primary breast cancer correlated with poor prognosis. The restoration of RORα expression in cancer cells suppresses cancer progression in 3D cultures and in mouse models [20]. These findings suggest that RORα is a novel, potent tumor suppressor.

RORα deficient mice suffer from immune abnormalities and display a stronger inflammatory response compared to wild-type mice [22]. The NF-κB pathway is a critical regulator of cytokine expression and inflammation [23,24]. The primary regulation of the NF-κB pathway is through association of NF-κB complexes with their inhibitor, I kappa B proteins. I kappa B binds to p65 or p50 subunit and inhibits their nuclear translocation. It has been shown that RORα negatively regulates the NF-κB signaling pathway by reducing p65 nuclear translocation [25,26]. This evidence suggests the potential function of RORα in inflammation; however, the exact roles of RORα in breast cancer-associated inflammation and macrophage infiltration remain to be determined.

In the present study, we showed that RORα expression inhibited ROS generation, cancer-associated macrophage infiltration, and breast cancer metastasis. We also identified complex I genes as RORα targets that mediate its function in regulating redox balance. These findings reveal a novel link between RORα downregulation, ROS generation, and cancer progression.

## 2. Results

### 2.1. RORα Inhibits Activation of Inflammation Gene Signature and Expression of Cytokines in Breast Cancer Cells

By comparing gene expression profiles between polarized and non-polarized mammary epithelial organoids, we identified RORα as a major regulator of mammary tissue polarity with inhibitory activity on tumor invasion [20]. Invasive breast cancer cell lines, including MDA-MB-231, MDA-MB-157, and BT-549, form the aggressive branching structures in 3D culture [20]. We found that the expression of RORα in these breast cancer cell lines significantly reduced the number of invasive branches and suppressed invasive tumor growth in 3D culture (Figure 1A and Appendix A).

To determine how RORα suppresses breast cancer progression, we performed gene expression profiling analysis in MDA-MB-231 and MDA-MB-157 cells with Affymetrix Exon Array, and identified many genes being differentially expressed in control and RORα-expressing cells (Figure 1B and Appendix A). The gene set enrichment analysis (GSEA) showed that the inflammatory pathway and expression of cytokine genes were inhibited by RORα in 3D culture (Figure 1C,D). Real-time RT-PCR data confirmed the results from microarray (Figure 1E). We also found that silencing RORα in S1 cells, a non-malignant mammary epithelial cell line, enhanced cytokine expression (Figure 1F). To further determine whether RORα inhibits cytokine protein expression and secretion, we quantified IL-6 levels in the conditioned medium collected from RORα-silenced and RORα-expressing cells with ELISA assay. Silencing RORα in S1 cells and MCF-10A cells increased the IL-6 protein levels in the conditioned medium, while overexpression of RORα decreased the level in breast cancer cells (Figure 1G and Appendix A). These results indicate that RORα expression suppresses the expression of the inflammation gene signature and cytokines in mammary epithelial cells.

### 2.2. RORα Inhibits ROS Production by Repressing the Expression of Complex I Genes

We previously showed that ROS production was elevated in non-polarized mammary epithelial cells, which in turn enhanced cytokine expression and monocyte infiltration in 3D culture [9]. Since silencing RORα induces the disruption of mammary tissue polarity [20], we asked whether the downregulation of RORα induces cytokine expression by elevating ROS levels. The knockdown of RORα significantly elevated ROS levels in non-malignant mammary epithelial cells, while introducing RORα in breast cancer cells reduced ROS production (Figure 2A and Appendix A). Notably, reducing ROS levels in the RORα-silenced MCF-10A cells with antioxidant *n*-Acetyl-L-cysteine (NAC) treatment suppressed cytokine gene expression (Figure 2B). These results indicate that ROS mediates RORα function in repressing the expression of cytokines and the inflammation gene signature.

To determine how RORα regulates ROS generation and cytokine expression, we performed gene co-expression analysis using the breast cancer TCGA dataset. We showed that the expression of mitochondrial complex I genes was negatively associated with RORα levels in human breast cancer tissues (Figure 2C). Real-time RT-PCR data showed that the introduction of RORα in MDA-MB-231 cells significantly inhibited the expression of multiple complex I genes (Figure 2D and Appendix A), while silencing RORα elevated the mRNA levels of these genes in MCF-10A cells (Figure 2E).

It was shown that electron leakage from complex I is one of the major sources of ROS [14,15]. We asked whether RORα suppresses ROS generation by repressing complex I gene expression. We performed a focused screening and found that silencing NDUFS6 or NDUFA11 significantly decreased ROS levels in MDA-MB 231-cells (Figure 2F and Appendix A). To further determine whether these two genes mediate RORα function in regulating ROS, we knocked down NDUFS6 and NDUFA11 expression in RORα-silenced MCF-10A cells. Reducing NDUFS6 or NDUFA11 expression partially suppressed ROS generation induced by RORα silencing (Figure 2G). The knockdown of NDUFS6 or NDUFA11 also repressed the expression of IL-6, IL-8, CXCL1, and CXCL3 in RORα-silenced MCF-10A cells (Figure 2H). These results indicate that RORα suppresses ROS generation and cytokine expression, in part, by repressing the expression of complex I genes.

Electron leakage from the respiration chain increases superoxide anion production [27]. To determine whether the RORα/complex I axis inhibits electron leakage in mitochondria, we measured superoxide anion levels with mitoSOX red in control and RORα-expressing breast cancer cells [28]. Silencing RORα significantly increased mitochondrial superoxide levels in MCF-10A cells, while expression of RORα in MDA-MB-231 cells reduced the superoxide accumulation (Figure 3A). We also found that silencing NDUFS6 or NDUFA11 significantly decreased superoxide levels in MDA-MB-231 cells (Figure 3B). Notably, reducing NDUFS6 or NDUFA11 expression suppressed the superoxide generation in RORα-silenced MCF-10A cells (Figure 3C).

The electron leakage in mitochondria is associated with oxygen consumption [15,29]. The oxygen consumption rate (OCR) can be quantified in real-time using the Seahorse XF96 Extracellular Flux Analyzer. Seahorse analysis showed that silencing RORα significantly reduced OCR in non-malignant mammary epithelial cells (Figure 3D,E). In contrast, the overexpression of RORα in breast cancer cells elevated OCR (Figure 3F). We also found that silencing NDUFS6 and NDUFA11 increased OCR in breast cancer cells (Appendix A). These results suggest that the inactivation of RORα in breast cancer cells enhances electron leakage in mitochondria by increasing NDUFS6 and NDUFA11 expression.

To determine how RORα suppresses the expression of complex I genes, we analyzed the sequence of NDAFS6 and NDAUFA11 genes, then identified several potential RORE in their promoter regions (Figure 4A). ChIP data showed that RORα bound to the promoter regions of NDUFS6 and NDUFA11 genes (Figure 4B). We also found that the expression of RORα negatively correlated with NDUFS6 and NDUFA11 mRNA levels in human breast cancer tissue (TCGA breast cancer dataset) (Figure 4C). These data suggest that RORα binds to RORE in the promoter region of NDUFS6 and NDUFA11, and suppresses the gene transcription.

### 2.3. RORα Expression Inhibits Mammary Tumor Metastasis and Macrophage Infiltration in Tumor Tissue

We showed previously that reduced RORα expression was associated with poor prognosis in breast cancer patients [20]. To determine how RORα expression is regulated during breast cancer development, we analyzed DNA copy number at RORA gene loci. The loss of RORA gene copy number was detected in 30% of human breast cancer tissue and is associated with reduced gene transcription (Figure 5A). Notably, low levels of RORα mRNA in breast cancer tissue are associated with short metastasis-free survival (Figure 5B).

To determine whether RORα expression suppresses breast cancer metastasis, we injected control and RORα-expressing MDA-MB-231-luc cells (luciferase-labeled) in SCID mice via the tail vein. RORα expression significantly inhibited MDA-MB-231 cell colonization in the lung (Figure 5C and Appendix A). The orthotopic mammary tumor model in immune competent mice is more physiologically relevant for studying cancer metastasis. Therefore, we further examined the roles of RORα in regulating cancer metastasis with the 4T1 model. The control and RORα-expressing 4T1 cells were injected into the BALB/c mice at the fourth mammary fat pad. Once tumors reached 1000 mm^3^, they were surgically removed. After three weeks, metastatic lesions in the lung were quantified. We found that the expression of RORα also significantly inhibited the lung metastasis of 4T1 cells in BALB/c mice (Figure 5D).

Data from gene expression profiling showed that RORα inhibited the inflammatory pathway (Figure 1C). The inflammation in the mammary tumor tissue is associated with macrophage infiltration and differentiation. We wondered whether RORα regulates macrophage accumulation in 4T1 xenografts. Immunohistochemistry analysis showed that F4/80 positive cells were reduced in RORα-expressing 4T1 tumor sections (Figure 6A). Quantified data from FACS analysis showed that RORα expression significantly reduced the number of total macrophages in 4T1 tumor tissues; interestingly, the portion of M1-like macrophages was increased upon RORα expression (Figure 6B–D, Appendix A). It was shown that M1-like macrophages suppress tumor progression and enhance immune response [30]. These results suggest that RORα suppresses breast cancer metastasis at least partially by reducing macrophage accumulation and by enhancing M1-like differentiation.

Next, we utilized a co-cultured system to determine how RORα inhibits macrophage infiltration. Control and RORα-expressing MDA-MB-231 cells were cultured in the lower chamber of a Transwell plate; THP-1 or differentiated THP-1 cells (M0) were plated in the upper chamber. We found that expressing RORα in breast cancer cells significantly decreased the number of THP-1 or differentiated THP-1 cells that migrated into the lower chamber (Figure 7A). Interestingly, antioxidant NAC treatment suppressed the THP-1 cell migration induced by RORα-silenced MCF-10A cells (Figure 7B). These results imply that RORα inhibits macrophage infiltration by reducing ROS levels.

Among the RORα-repressed cytokines, IL-6 is a pro-inflammatory factor that induces macrophage/monocyte infiltration and differentiation [31,32]. We found that the addition of IL-6 rescued the THP-1 cell migration that was suppressed by RORα- expressing MDA-MB-231 cells in the co-culture assay (Figure 7C). These results suggest that the function of RORα in suppressing macrophage infiltration is at least partially mediated by IL-6.

## 3. Discussion

RORα was identified as a potential tumor suppressor in breast cancer [33]; however, the molecular and cellular mechanisms by which it suppresses cancer progression are not fully understood. In the present study, we showed that the downregulation of RORα in breast cancer cells induced ROS generation by enhancing complex I gene expression. The increased ROS levels in mammary epithelial cells is sufficient to induce cytokine expression and macrophage migration. We also found that RORα expression inhibited cancer-associated macrophage infiltration and breast cancer metastasis in vivo. These results identified new roles of RORα in repressing cancer metastasis and ROS-associated inflammation.

Macrophage infiltration is detected at the early stage of breast cancer development [5,6,34]. Macrophages also accumulate around the terminal end buds of mammary glands rather than the polarized ductal epithelial cells [35,36]. One common feature shared by the terminal end buds and the mammary tumor is the presence of multilayer non-polarized epithelial cells [37,38]. Using 3D culture assay, we further confirmed the association between the disruption of tissue polarity and macrophage infiltration and identified ROS as a potential molecule link between mammary tissue polarity and macrophage infiltration [9]. Reduced RORα expression is detected in non-polarized breast cancer cells and terminal end buds [20,39]. In addition, RORα expression is sufficient to restore tissue polarity in malignant mammary epithelial cells [20]. We showed that RORα suppressed ROS generation and subsequently inhibited cytokine expression and macrophage infiltration. These results imply that RORα is the major regulator of ROS production and inflammation associated with the disruption of tissue polarity (Figure 7D).

It is well established that the mitochondria respiration chain generates significant amounts of superoxide [40]. Superoxide reacts with manganese SOD (MnSOD) in the mitochondrial matrix to generate H2O2, which can cross the mitochondrial outer membrane to access cytosolic targets [41]. A study in the nonalcoholic steatohepatitis model showed that RORα reduced ROS levels in hepatocytes and suppressed hepatic oxidative stress, which is accompanied with induction SOD2 and Gpx1 [42]. We showed that RORα expression was negatively associated with mRNA levels of complex I genes in human breast cancer tissues. Using ChIP analysis and functional rescue experiments, we demonstrate that RORα suppressed ROS generation by directly repressing the expression of NDUFS6 and NDUFA11. The negative correlation between RORα expression and mRNA levels of NDUFS6 and NDUFA11 suggests that these two genes are the major targets of RORα mediating its function in ROS regulation during breast cancer progression.

It was reported that inhibiting the activity of complex I enhances the ROS production and promotes cancer cells’ migration and invasion [43]. Mammalian complex I contains 45 subunits, and the roles of complex I subunits in ROS generation are more complicated. We showed that silencing NDUFS6 and NDUFA11 suppressed superoxide and ROS generation in breast cancer cells. In addition, the upregulation of complex I subunits in RORα-silenced mammary epithelial cells was accompanied by reduced OCR, while silencing NDUFS6 and NDUFA11 increased OCR in breast cancer cells (Appendix A). These results imply that an increased expression of NDUFS6 and NDUFA11 in mammary epithelial cells enhances the electron leakage in mitochondria. A study in neurons and astrocytes provides a possible explanation as to the different functions of complex I in ROS generation. Mitochondrial complex I in neurons mainly embeds in supercomplexes with highly efficient mitochondrial respiration and low ROS production. However, the abundance of free complex I is higher in astrocytes, which leads to the significantly high ROS production [44]. These data suggest that the complex I status dictates its roles in ROS production. It would be interesting to determine whether the reduction of RORα in cancer cells leads to the accumulation of free complex I in the future.

Increased ROS production induces the activation of redox-sensitive transcription factors, such as HIF-1α and NF-κB, expression of pro-inflammatory cytokines, and release of inflammasomes. Mitochondrial-derived oxidative stress is associated with chronic inflammation and cancer progression. Mitochondrial-derived ROS also contributes to the production of pro-inflammatory cytokines IL-1β, IL-6, and TNF-α [41]. It was reported that RORα suppresses the NF-κB pathway and inflammation [45]. Our data suggest that ROS is the major mediator of RORα function in regulating cytokine expression and cancer-associated macrophage infiltration. Cancer-associated macrophages create an inflammatory environment to stimulate angiogenesis, promote tumor growth, and enhance tumor cell migration and invasion [46,47]. IL-6 is an important cytokine that regulates macrophage infiltration and differentiation; it can also promote breast cancer metastasis by enhancing epithelial–mesenchymal transition [32,48]. We showed that IL-6 is one of the cytokines suppressed by RORα in mammary epithelial cells. The results from the co-culture experiments suggest that RORα inhibits macrophage accumulation in breast cancer tissue at least partially through IL-6 repression.

In summary, our data reveal the new function of RORα in repressing ROS generation by inhibiting the expression of complex I genes. Restoring RORα expression suppresses cancer metastasis and ROS-associated inflammation; therefore, enhancing RORα expression or inducing RORα activation with agonists is a potential strategy to suppress ROS-associated cancer progression.

## 4. Materials and Methods

### 4.1. Antibodies and Reagents

Matrigel was purchased from BD Bioscience (Bedford, MA, USA). ShRNA constructs selectively targeting NDUFS6, NDUFA7, NDUFA9, and NDUFA11 were purchased from Sigma Aldrich (MISSION shRNA library, St. Louis, MO, USA). Recombinant human IL-6 was purchased from Peprotech (Cranbury, NJ, USA; 200-06). NAC (A7250) was purchased from Sigma Aldrich (St. Louis, MO, USA). Antibody CD16/32 (101302, clone:93) was purchased from BioLegend (San Diego, CA, USA). Antibodies F4/80 (1950719, clone: BM8), CD11c (2011154, clone: N418), CD45 (2055168, clone: 104), CD11b (2011193, clone: M1/70), CD86 (1987724, clone: GL1), and CD206 (2073756, clone: MR6F3) were purchased from Invitrogen (Waltham, MA, USA). Antibody RORα (E0713) was purchased from Santa Cruz Biotechnology (Dallas, TX, USA). Antibodies Flag (F1804) and tubulin (AB9354) were purchased from Sigma Aldrich (St. Louis, MO, USA). Antibody NDUFA11 (17879-1-AP) was purchased from Proteintech.

### 4.2. Cell Lines and Culture Conditions

The MDA-MB-231 cells were cultured in DMEM/F12 (Sigma Aldrich, St. Louis, MO, USA; D8437) with 10% FBS (Sigma Aldrich, St. Louis, MO, USA; F2442) and 1% Pen/Strep (Sigma Aldrich, St. Louis, MO, USA; P4333). BT549 cells were cultured in RPMI-1640 (Sigma Aldrich, St. Louis, MO, USA; R8758) with 10% FBS and 1% Pen/Strep. MDA-MB-157 cells were cultured in DMEM (Sigma Aldrich, St. Louis, MO, USA; D6429) with 10% FBS and 1% Pen/Strep. The HMT-3522 S1 cells were cultured in DMEM/F12 with 250 ng/mL insulin, 10 μg/mL transferrin, 2.6 ng/mL sodium selenite, 10^−10^ M β-estradiol, 1.4 μM hydrocortisone, 5 μg/mL prolactin, and 10 ng/mL EGF. MCF-10A cells were cultured in DMEM/F12 with 5% horse serum, 20 ng/mL EGF, 0.5 mg/mL hydrocortisone, 100 ng/mL cholera toxin, 10 μg/mL insulin, and 1% Pen/Strep. THP-1 cells were cultured in RPMI-1640 with 10% FBS, 1% Pen/Strep, and 0.05 mM 2-mercaptoethanol. HEK293 FT cells (A kind gift from Dr. Mina J Bissell, Lawrence Berkeley National Laboratory) were cultured in DMEM with 10% FBS, 0.1 mM Non-Essential Amino Acids (Sigma Aldrich, St. Louis, MO, USA; M7145), 6 mM L-glutamine (VWR, Atlanta, GA, USA; 20J1956675), 1 mM Sodium Pyruvate (Sigma Aldrich, St. Louis, MO, USA; S8636), and 1% Pen/Strep. The cells were grown in a humidified incubator at 37℃ with 5% CO_2_. All the cells were tested for mycoplasma contamination every two months.

### 4.3. 3D Culture

A three-dimensional lrECM on-top culture was performed as previously described [9]. 150 ul of Matrigel (Corning, NY, USA; 6347014) was plated on the bottom of a 24-well plate; MDA-MB-231 cells, MDA-MB-157 cells, and BT-549 cells (0.4 × 10^5^ cells per well) were seeded on the top of the Matrigel layer, and the additional medium containing 10% Matrigel was added on the top.

### 4.4. Co-Culture Assay

After MDA-MB-231 cells and MCF-10A cells were cultured in a 24-well plate for 24 h, R18-dyed THP-1 cells or differentiated THP-1 cells (M0 macrophage) were seeded to a transwell plate (Thermo Fisher Scientific, Rockford, IL, USA; 140629). THP-1 cells were differentiated to M0 macrophages with 150 nM PMA (Sigma Aldrich, St. Louis, MO, USA; P1585) treatment. Fluorescence images were taken 24 h later with an Eclipse 80i microscope (Nikon, Tokyo, Japan). The pictures were taken under fixed exposure conditions.

### 4.5. Microarray Analysis and Quantitative RT-PCR

Control and RORα-expressing MDA-MB 231 and MDA-MB 157 cells were isolated from 3D cultures as previously described [39]. Total cellular RNA was extracted using an RNeasy minikit with on-column DNase digestion (Qiagen, Hilden, Germany). Affymetrix microarray analysis was performed using the Affymetrix HuGene-1.0 highthroughput array (HTA) GeneChip system. Preprocessing, normalization, and filtering were performed using the R Bioconductor. The R package limma [49] was used to perform differential expression analysis comparing control and RORα-expressing for each of the two cell lines, 157 and 231, separately. Significant differentially expressed genes were determined as false discovery rate q-value < 0.20. A heatmap was generated for results visualization. Gene set enrichment analysis was performed with GSEA v2.07 [50].

Complementary DNA was synthesized using the SuperScript^TM^ III First-Strand Synthesis kit (Invitrogen, Waltham, MA, USA; 18-080-051) from 1 μg RNA samples. Quantitative RT-PCR was carried out with SYBR Green PCR Master Mix reagents using an StepOnePlus^TM^ Real-Time PCR System (Thermo Fisher Scientific, Rockford, IL, USA; 4376600). Thermal cycling was conducted at 95 °C for 30 s, followed by 40 cycles of 5 s amplification at 95 °C, 55 °C for 30 s, and 72 °C for 15 s. The relative quantification of gene expression for each sample was analyzed by the threshold cycle (CT) method. Information on the primers used for the amplification of RORα, IL-6, IL-8, IL-24, IL-32, Cxcl1, Cxcl2, Cxcl3, Cxcl5, Cxcl10, NDUFS6, NDUFA7, NDUFA9, NDUFA11, and 18S ribosomal RNA is given in Appendix A.

### 4.6. Seahorse Assay

MDA-MB-231 cells (shcontrol, shNDUFS6, shNDUFA11), S1 cells (shcontrol, shRORα), and MDA-MB-157 cells (control, RORα-expressing) were seeded to a seahorse XF96 microplate (10,000 cells per well). After 24 h, the cell culture microplate was placed into a 37 °C non-CO_2_ incubator for 1 h. Then, the mitochondrial respiration was measured in the XF Analyzer according to the Agilent Seahorse XF96 Cell Mito Stress Test Assay manual.

### 4.7. ROS and Superoxide FACS Analysis

According to the CellROX Deep Red Flow Cytometry Assay Kit manual (Invitrogen, Waltham, MA, USA; C10491), cells were trypsinized and resuspended at a concentration of 0.5 × 10^5^ cells/mL. The Deep Red reagent was added to the cell samples at 1 μM and incubated for 45 min at 37 °C and protected from light. One microliter propidium iodide (1 mg/mL) was added to the cell samples and incubated on ice for FACS analysis.

According to the MitoSOX^TM^ Red manual (Invitrogen, Waltham, MA, USA; M36008), cells were trypsinized and resuspended at a concentration of 0.5 × 10^5^ cells/mL. The MitoSOX^TM^ reagent was added to the cell samples at 5 μM and incubated for 30 min at 37 °C and protected from light. Then cells were resuspended in PBS for FACS analysis. FACS analysis was done with Becton Dickinson LSR II (San Diego, CA, USA) and data were analyzed by FlowJo.

### 4.8. Chromatin Immunoprecipitation (ChIP) Assay

Flag-tagged RORA cDNA were cloned into pCDH1 plasmid and generated expression vector pCDH1-RORα-Flag. HEK293 FT cells were transfected with pCDH1 or pCDH1-RORα-Flag plus packaging lentivector using FuGENE (Promega, Madison, WI, USA; 0000356676). MDA-MB-231 cells were infected with lentivirus and selected by puromycin 48 h after infection. Vector control and RORα-expressing MDA-MB-231 cells were cross-linked using formaldehyde for the ChIP assay. The ChIP assay was performed based on the Upstate Biotechnology ChIP protocol, with a few modifications [51]. After formaldehyde cross-linking, nuclei were isolated with a nuclear isolation kit (Sigma Aldrich, St. Louis, MO, USA) and resuspended in ChIP lysis buffer (1% SDS, 10 mM EDTA, 50 mMTris-HCl [pH8.0]) containing protease inhibitor cocktail. Protein-DNA complexes were immunoprecipitated as per the Upstate protocol. Ten percent of the extracted chromatin was aliquoted as input, and 5% of input was used as a template for qPCR. Isolated DNA was then analyzed by quantitative PCR using the following primers: NDUFS6 promoter, 5′- AAGGTTTCGCACACCATTGC-3′and 5′-GATTCAGGTGGTCACCCGTT-3′, and NDUFA11 promoter, 5′-GCTATGGCTCCCAATGCCTA-3′ and 5′- CGTGTGCACTTGTATAGACGC-3′.

### 4.9. Mouse Experiments

For the xenograft experiments, 6-week-old female BALB/c mice were randomly grouped and injected with 1 × 10^6^ control or RORα-expressing 4T1 cells at the 4th mammary fat pad. The tumors were measured with a caliper every other day. The tumor volume (mm^3^) was estimated using the formula [volume = π × (width)^2^ × (length) /6]. Once tumors reached 1000 mm^3^ (25 days after tumor cell implantation), the primary tumors were removed by surgery. Xenograft tumor sections were de-paraffinized and rehydrated. An immunohistochemistry analysis of F4/80 positive cells was performed as described previously [52]. Three weeks after primary tumor removal, the lung tissue was collected for fixation and hematoxylin and eosin (HE) staining for the detection of metastases in the lungs. Images were taken with a Nikon microscope.

For the lung colonization experiments, 6-week-old female severe combined immunodeficient (SCID) mice were randomly grouped and injected with 1 × 10^6^ control or RORα-expressing MDA-MB-231-luc cells via the tail vein. To detect lung metastasis, bioluminescent images were taken every week after injection using in vivo imaging system (IVIS). All mouse experiments were approved by the University of Kentucky Institutional Animal Care and Use Committee.

### 4.10. Tumor Infiltrating Lymphocytes Isolation and FACS Analysis

Primary tumor tissues from control and RORα-expressing 4T1 xenografts were digested with collagenase (Sigma Aldrich, St. Louis, MO, USA; C6885). Single cells were isolated and blocked by CD16/32 for 10 min and stained by cell surface markers F4/80, CD11c, CD45, CD11b, CD86, and CD206 for 30 min at 4 °C, and protected from light. The cells were then resuspended for FACS analysis. Macrophages were identified as DAPI-, CD45+, CD11b+, CD11c-, and F4/80+. The macrophages were then gated for M1 (CD86+CD206-) and M2 (CD86- CD206+) [53].

### 4.11. Kaplan-Meier Survival Analysis and Other Statistical Analysis

The clinical relevance of reduced RORα expression was assessed by analyzing mRNA levels of RORα and patient survival using the published microarray data generated from 6,365 human breast cancer tissue samples [54]. Significant differences in distant metastasis-free survival time were assessed with the Kaplan–Meier survival analysis and the Cox proportional hazard (log-rank) test.

All experiments were repeated at least twice. Inferential statistics were used to compare data sets from different experimental groups and reported data were the mean ± standard error of mean (SEM). Student’s *t*-tests (two groups) and one-way ANOVA (three or more groups) were used to determine the significant differences between means and were performed with Prism 6.01 (Graph Pad Software, San Diego, CA, USA). The minimum statistical significance was set at *p* < 0.05.

## Figures and Tables

**Figure 1 ijms-22-10665-f001:**
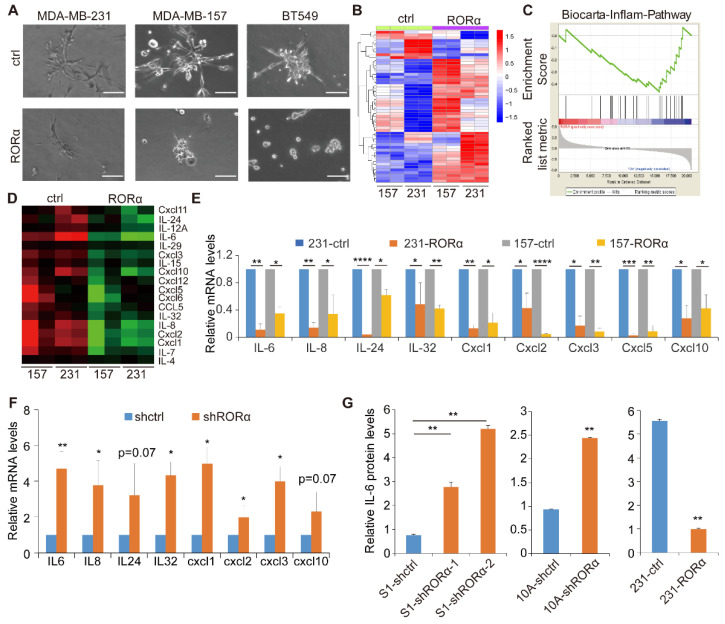
RORα expression suppresses cytokine gene expression in mammary epithelial cells. (**A**) Phase images showing control and RORα-expressing MDA-MB-231, MDA-MB-157, and BT-549 cells in 3D culture after being plated for 4 days. Bar: 100 µm. (**B**) Heatmap showing different gene expression profiles in control and RORα-expressing MDA-MB-231 and MDA-MB-157 cells. (**C**) Gene set enrichment analysis (GSEA) data showed that the inflammatory pathway was inhibited by RORα in MDA-MB-231 and MDA-MB-157 cells in 3D culture. (**D**) Heatmap showing cytokine gene expression in control and RORα-expressing MDA-MB-231 cells in 3D culture. (**E**) Real-time RT-PCR quantified mRNA levels of cytokines genes in control and RORα-expressing MDA-MB-231 and MDA-MB-157 cells; results are presented as mean ± SEM; *n* = 4, **** *p* < 0.0001, *** *p* < 0.001, ** *p* < 0.01, * *p* < 0.05, one-way ANOVA test. (**F**) Real-time RT-PCR quantified cytokines gene expression in control and RORα-silenced S1 cells; results are presented as mean ± SEM; *n* = 4, ** *p* <0.01, * *p* <0.05, student’s *t*-tests. (**G**) ELISA analysis quantified the protein levels of IL-6 in the conditioned medium collected from control and RORα-silenced S1 and MCF-10A cells, and from control and RORα-expressing MDA-MB-231 cells; results are presented as mean ± SEM; *n* = 3, ** *p* < 0.01, * *p* < 0.05, one-way ANOVA test.

**Figure 2 ijms-22-10665-f002:**
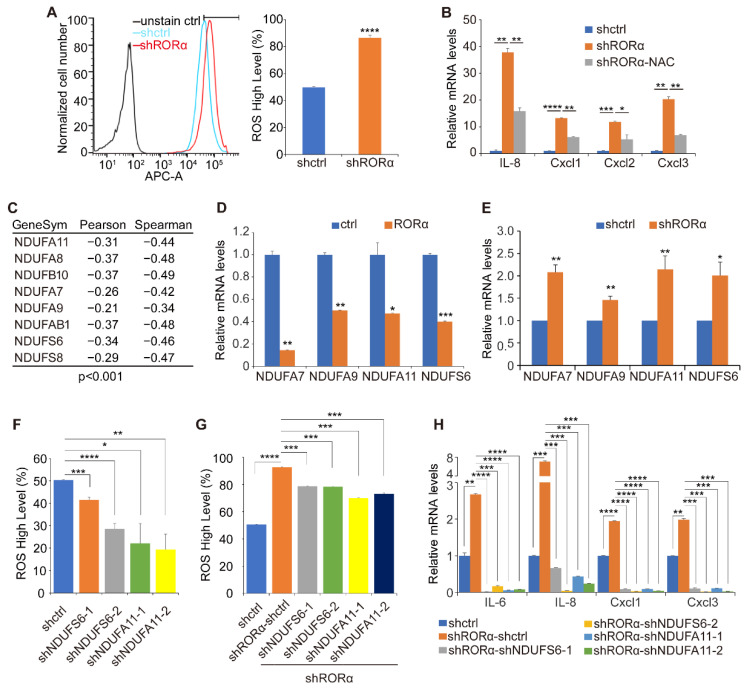
ROS mediates RORα function in repressing cytokine expression. (**A**) FACS analysis quantified ROS levels in control and RORα-silenced MCF-10A cells, the top 50% of cell population in control group was gated as “ROS High”; results are presented as mean ± SEM; *n* = 4, **** *p* < 0.0001, student’s *t*-tests. (**B**) Real-time RT-PCR quantified mRNA levels of cytokines genes in RORα-silenced MCF-10A cells cultured with or without NAC (4 μM) treatment for 48h; results are presented as mean ± SEM; *n* = 4, **** *p* < 0.0001, *** *p* < 0.001, ** *p* < 0.01, * *p* < 0.05, one-way ANOVA test. (**C**) Table showing the significantly negative correlation between RORα and complex I gene mRNA levels in human breast cancer tissue, gene expression data were derived from the TCGA breast cancer dataset. (**D**) Real-time RT-PCR quantified mRNA levels of mitochondrial complex I genes in control and RORα-expressing MDA-MB-231 cells; results are presented as mean ± SEM; *n* = 4, *** *p* < 0.001, ** *p* < 0.01, * *p* < 0.05, student’s *t*-tests. (**E**) Real-time RT-PCR quantified mRNA levels of mitochondrial complex I genes in control and RORα-silenced MCF-10A cells; results are presented as mean ± SEM; *n* = 4, ** *p* < 0.01, * *p* < 0.05, student’s *t*-tests. (**F**) FACS quantification of ROS levels in control or NDUFS6-silenced or NDUFA11-silenced MDA-MB-231 cells; results are presented as mean ± SEM; *n* = 4, **** *p* < 0.0001, *** *p* < 0.001, ** *p* < 0.01, * *p* < 0.05, one-way ANOVA test. (**G**) FACS quantification of ROS levels in control or RORα-silenced or RORα&NDUFS6-silenced or RORα&NDUFA11-silenced MCF-10A cells; results are presented as mean ± SEM; *n* = 4, **** *p* < 0.0001, *** *p* < 0.001, one-way ANOVA test. (**H**) Real-time RT-PCR quantified mRNA levels of cytokines genes; NDUFS6 or NDUFA11 knockdown partially rescued the expression of these genes in RORα-silenced MCF-10A cells; results are presented as mean ± SEM; *n* = 4, **** *p* < 0.0001, *** *p* < 0.001, ** *p* < 0.01, one-way ANOVA test.

**Figure 3 ijms-22-10665-f003:**
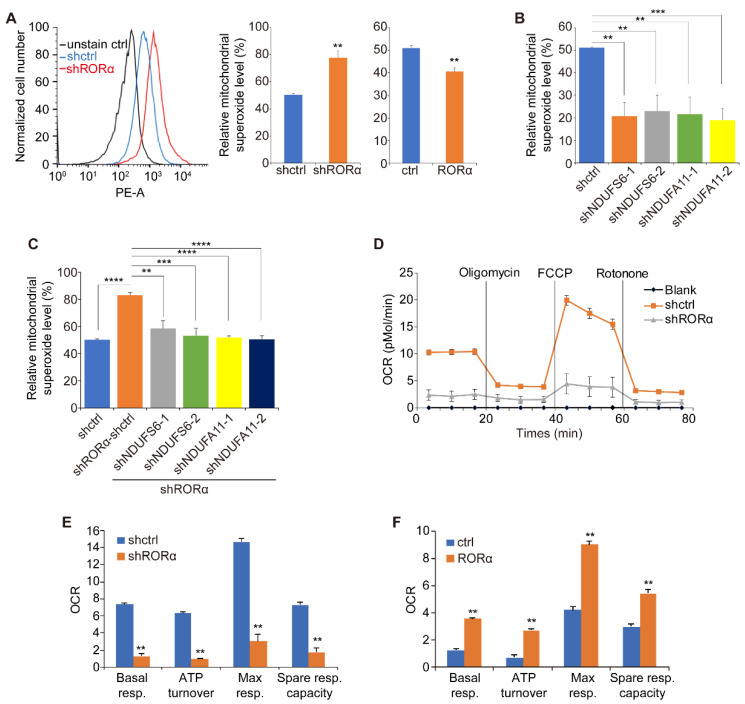
RORα inhibits ROS production by repressing the expression of complex I genes. (**A**) FACS quantification of mitochondrial superoxide levels in control and RORα-silenced MCF-10A cells, and in control and RORα-expressing MDA-MB-231 cells; results are presented as mean ± SEM; *n* = 4, ** *p* < 0.01, student’s *t*-tests. (**B**) FACS quantification of mitochondrial superoxide levels in control or NDUFS6-silenced or NDUFA11-silenced MDA-MB-231 cells; results are presented as mean ± SEM; *n* = 4, *** *p* < 0.001, ** *p* < 0.01, one-way ANOVA test. (**C**) FACS quantification of mitochondrial superoxide levels in control or RORα-silenced or RORα&NDUFS6-silenced or RORα&NDUFA11-silenced MCF-10A cells; results are presented as mean ± SEM; *n* = 6, **** *p* < 0.0001, *** *p* < 0.001, ** *p*< 0.01, one-way ANOVA test. (**D**) Seahorse analysis quantified OCR in control and RORα-silenced S1 cells. (**E**) Seahorse analysis quantified OCR in control and RORα-silenced S1 cells; results are presented as mean ± SEM; *n* = 12, ** *p* < 0.01, student’s *t*-tests. (**F**) Seahorse analysis quantified OCR in control and RORα-expressing MDA-MB-157 cells; results are presented as mean ± SEM; *n* = 12, ** *p* < 0.01, student’s *t*-tests.

**Figure 4 ijms-22-10665-f004:**
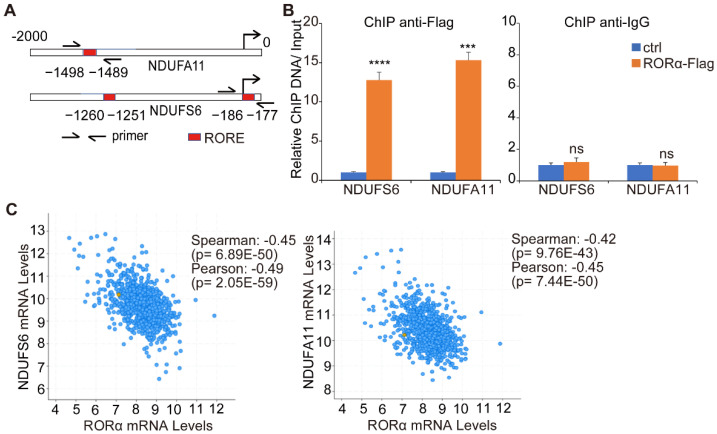
RORα binds to ROREs in NDUFS6 and NDUFA11 promoter regions. (**A**) Scheme showing the potential RORE in NDAFS6 and NDAUFA11 genes. The red box represents potential RORE in their ~2000 bp promoter regions. (**B**) Bar graph showing the relative ChIP DNA enrichment in control and RORα-overexpression MDA-MB-231 cells. After normalization to input, the relative enrichment of ChIP DNA was calculated in both groups by dividing with the value of the control group; results are presented as mean ± SEM; *n* = 4, **** *p* < 0.0001, *** *p* < 0.001, student’s *t*-tests. (**C**) Dot plots showing the negative correlation between mRNA levels of NDUFS6 or NDUFA11 and RORα expression in human breast cancer tissues (gene expression data were derived from the TCGA breast cancer dataset).

**Figure 5 ijms-22-10665-f005:**
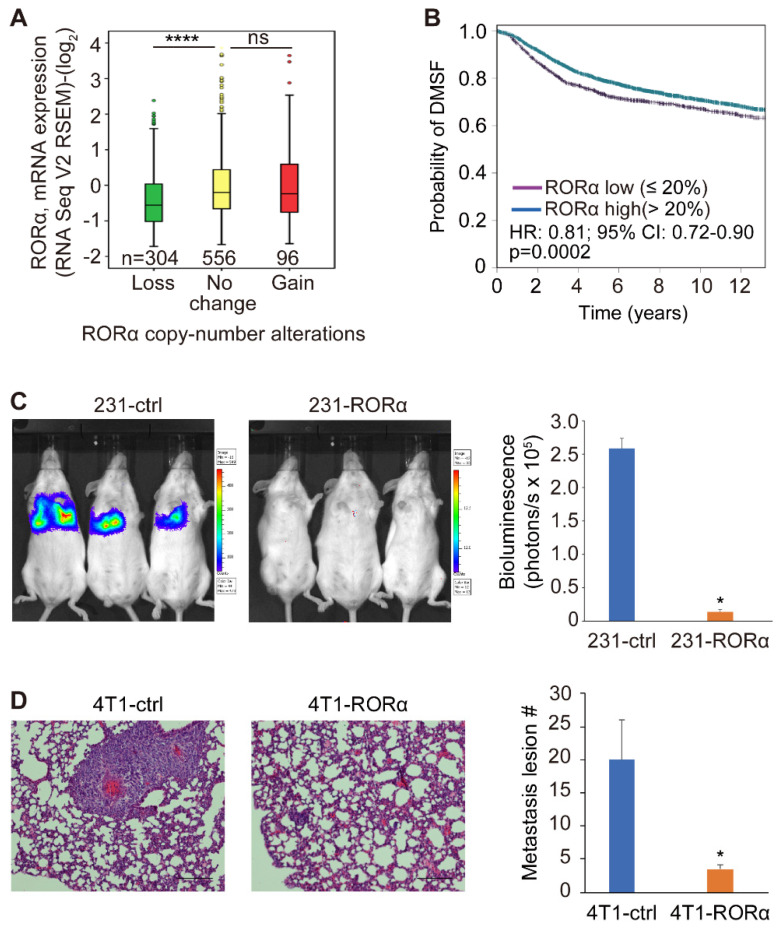
RORα expression inhibits breast cancer metastasis in mice. (**A**) Reduction of RORα expression was negatively associated with LOH in human breast cancer tissue, *n* = 860, **** *p* < 0.0001, one-way ANOVA test. (**B**) Kaplan-Meier survival analysis showing reduced RORα mRNA level is associated with short distant recurrence-free survival, *n* = 6365. (**C**) IVIS images (left) and quantification data (right) showing lung colonization of control and RORα-expressing MDA-MB-231-luc cells in SCID mice; results are presented as mean ± SEM; *n* = 6, * *p* < 0.05, student’s *t*-tests. (**D**) HE staining (left) and quantification (right) of metastatic lesions of 4T1 cells in the lung; results are presented as mean ± SEM; *n* = 4, * *p* < 0.05, student’s *t*-tests. Bar: 100 µm.

**Figure 6 ijms-22-10665-f006:**
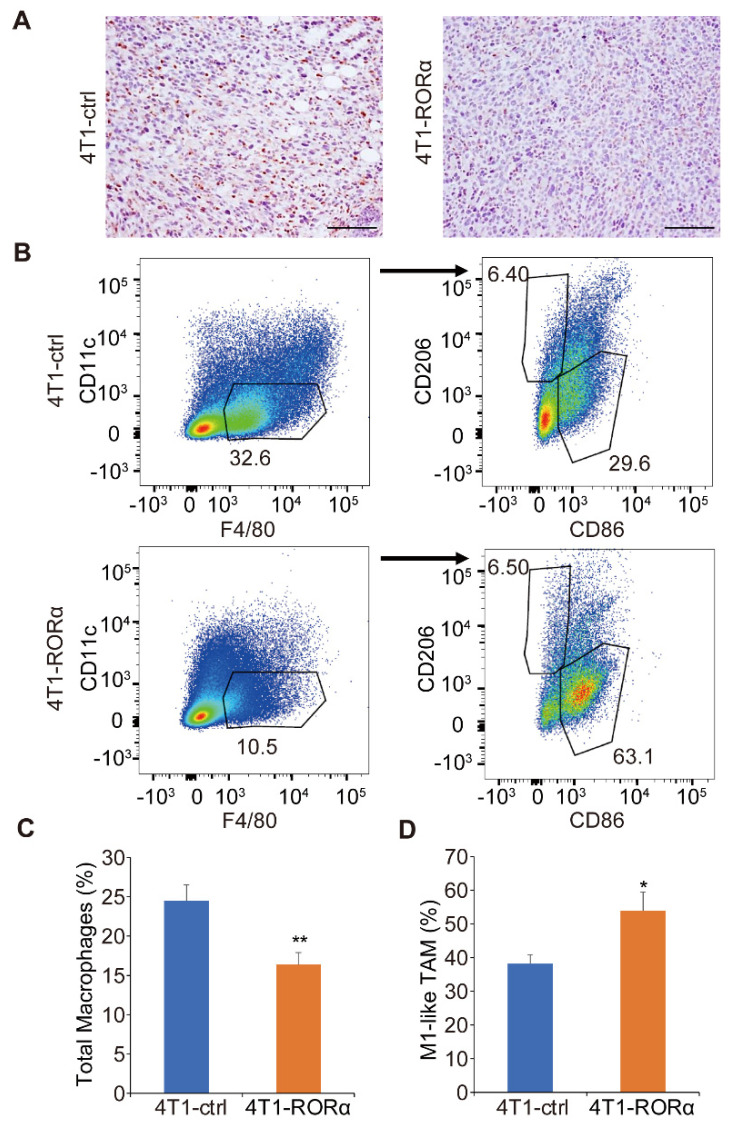
RORα inhibits macrophage accumulation and M2 polarization in mammary tumor tissue. (**A**) Images showing the accumulation of F4/80 positive cells in control and RORα-expressing 4T1 xenografts. Bar: 100 µm. (**B–D**) FACS analysis and quantification of macrophages isolated from control or RORα-expressing 4T1 cells injected BALB/c mice; results are presented as mean ± SEM; *n* = 7, ** *p* < 0.01, * *p* < 0.05, student’s *t*-tests.

**Figure 7 ijms-22-10665-f007:**
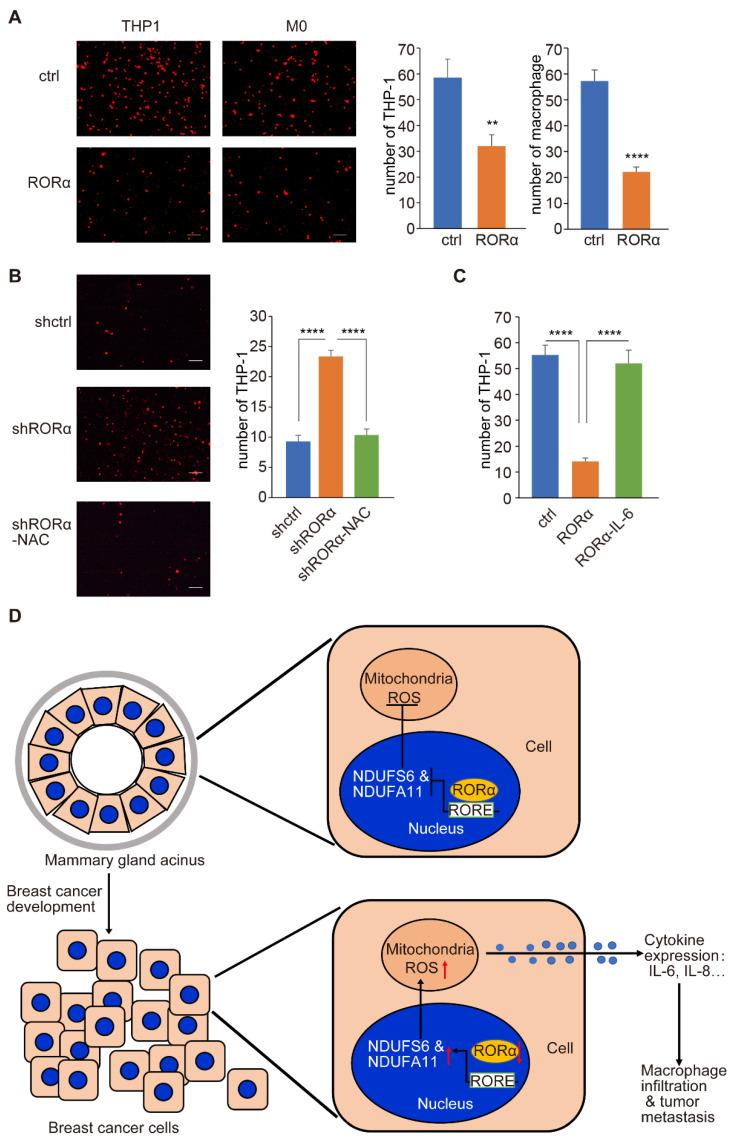
RORα expression in mammary epithelial cells inhibits the migration of THP-1 cells in the co-culture assay. (**A**) Images and quantification data showing the migration of THP-1 or differentiated THP-1 cells in the co-culture assay. THP-1 or differentiated THP-1 cells were cultured with control or RORα-expressing MDA-MB-231 cells (M0); results are presented as mean ± SEM; *n* = 4, **** *p* < 0.0001, ** *p* < 0.01, student’s *t*-tests. Bar: 100 µm. (**B**) Images and quantification data showing the migration of THP-1 cells in the co-culture assay. THP-1 cells were co-cultured with RORα-silenced MCF-10A cells treated with or without NAC (4 μM) for 24 h; results are presented as mean ± SEM; *n* = 4, **** *p* < 0.0001, student’s *t*-tests. Bar: 100 µm. (**C**) Quantification data showing the migration of THP-1 cells in the co-culture assay. THP-1 cells were co-cultured with RORα-expressing MDA-MB-231 cells in the presence or absence of IL-6 (20 ng/mL); results are presented as mean ± SEM; *n* = 4, **** *p* < 0.0001, one-way ANOVA test. (**D**) Scheme showing the molecular mechanism by which RORα suppresses ROS-associated macrophage accumulation and cancer progression.

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
