# Peer review of "RORα Suppresses Cancer-Associated Inflammation by Repressing Respiratory Complex I-Dependent ROS Generation"

_ijms, 2021, doi:10.3390/ijms221910665_

Round 1
Reviewer 1 Report
#1. Concerted efforts on the part of the authors is commendable and the manuscript is very well designed and the experiments have been conducted so as to prove the point of concept. #2. The authors may evaluate the levels of proteins upregulated or downregulated (western blotting) to analyze the proteins regulated by ROR-alpha in the different breast cancer cell lines utilized for other experiments.This will clearly delineated the expression of ROR-alpha under different conditions.#3. The authors have carried out xenograft experiments in immunocompromised mice, similarly, the breast carcinoma cells with upregulated and knock-out ROR-alpha can be sub-cutaneouly injected either in the mammarpad or sc under the skin on the dorsal side.The tumor growth can be documented by measuring the tumor size biweekly/weekly over a period of 4-6 weeks.The luciferase imaging of these sc tumors can also be done weekly.These experiments will further validate the results.
Author Response
#1. Concerted efforts on the part of the authors is commendable and the manuscript is very well designed and the experiments have been conducted so as to prove the point of concept.
Response: We thank the review for this and following comments.
#2. The authors may evaluate the levels of proteins upregulated or downregulated (western blotting) to analyze the proteins regulated by ROR-alpha in the different breast cancer cell lines utilized for other experiments. This will clearly delineated the expression of ROR-alpha under different conditions.
Response: Since complex I genes were regulated by RORa at the transcription level, we have examined mRNA levels of complex I genes in human breast cancer tissues. We showed that the mRNA levels were negatively correlated with RORa expression in human breast cancer tissue (Fig. 4C).
#3. The authors have carried out xenograft experiments in immunocompromised mice, similarly, the breast carcinoma cells with upregulated and knock-out ROR-alpha can be sub-cutaneouly injected either in the mammarpad or sc under the skin on the dorsal side.The tumor growth can be documented by measuring the tumor size biweekly/weekly over a period of 4-6 weeks.The luciferase imaging of these sc tumors can also be done weekly. These experiments will further validate the results.
Response: We now include IVIS weekly images and quantification data in Supplemental figure 3. Our previous study showed that RORα expression suppressed MDA-MB-231 and T4-2 xenograft growth (Cancer Res, 2012); similar results were also obtained in 4T1 xenograft. We did not include those data in this manuscript because RORa function in inhibiting tumor growth has been published.
Reviewer 2 Report
The manuscript by Moa W et al. titled as “RORα suppresses cancer-associated inflammation by repressing respiratory complex I-dependent ROS generation” examines the role of ‘Retinoid orphan nuclear receptor alpha’ (RORα), a member of the orphan nuclear receptor family, in the regulation of Reactive oxygen species (ROS) generation in breast cancer cells. Using in-vitro and in-vivo approaches authors showed that RORα expression inhibits the ROS induced cytokine expression (e.g., IL-6, IL-8) and associated macrophage accumulation and tumor metastasis. Also, the authors have highlighted significantly the interaction of RORα with mitochondrial complex I genes NDUFS6 and NDUFA11 inhibiting the ROS generation required for cancer progression. This article is well written, and the results presented are significant and have been interpreted appropriately. However, there are some issues that authors should consider clarifying
- The authors conclude that in the breast cancer cells ROS induced NF-kB mediating expression of pro-inflammatory cytokines like IL-6 etc. responsible for tumor progression, but there is no experimental evidence provided on the NF-kB role in this study.
- One significant finding of this study is RORα selective binding in the promoter region of mitochondrial complex I genes NDUFS6 and NDUFA11. However, it is not clear whether this interaction is specific for RORα isoforms in particular? Also, authors should include about RORα isoforms/structure to the introduction section.
Some minor points to be considered for correction accordingly-
- Fig.1B Heatmap gene names are not legible. It can be replaced with a better resolution image of the same.
- Correct the figures where labels are missing, e.g. In Fig2. ‘H’ is missing.
- In Fig.2 legend “C” should be reframed accordingly, it is not a dot plot.
- In Fig.4A: ‘-1500’ and the ‘red box’ represent what should be mentioned in the figure legend.
- In Materials and Methods section- line 399 “4.33.D” should be replaced with “4.3. 3D”.
- Antioxidant N-Acetyl-L-cysteine (NAC) treatment method (and the used concentration) should add in the material and methods section.
- All the reference numbers should be corrected wherever is needed. e.g. In line 554, Ref no. “20” is actually a DOI of the reference 19. Likewise, reference numbers 22, 24 etc. need to be corrected accordingly.
Author Response
1. This article is well written, and the results presented are significant and have been interpreted appropriately. However, there are some issues that authors should consider clarifying
Response: Thanks the review for this and following comments.
2. The authors conclude that in the breast cancer cells ROS induced NF-kB mediating expression of pro-inflammatory cytokines like IL-6 etc. responsible for tumor progression, but there is no experimental evidence provided on the NF-kB role in this study.
Response: Our previous study showed ROS elevation induced NF-kB in mammary epithelial cells (JCS, 2017). However, the review is right that we did not provide data related to NF-kB. Therefore, we modified the scheme in figure 7 and changed the conclusion to ‘RORα suppresses cancer-associated inflammation by repressing respiratory complex I-dependent ROS generation’.
3. One significant finding of this study is RORα selective binding in the promoter region of mitochondrial complex I genes NDUFS6 and NDUFA11. However, it is not clear whether this interaction is specific for RORα isoforms in particular? Also, authors should include about RORα isoforms/structure to the introduction section.
Response: We have included RORα isoform information in the introduction section on page 2. We showed previously that both RORα1 and 4 were expressed in mammary epithelial cells and had tumor suppressor function (Cancer Res, 2012). The current study focused on RORα1.
4. Some minor points to be considered for correction accordingly-
Response: We are sorry for these oversights. We have uploaded the new images and corrected the mistakes in figure legends, material and methods, and references.
Reviewer 3 Report
Overall link between ROR alfa and breast cancer metastasis presented by authors is interesting. However some major revisions should be done and additional data provided.
All figure legends and experimental procedures need to be more detailed. Figures are not self explanatory and therefore sometimes hard to interpret.
Figure 1: A) Images and structure of 3D spheroids does not look representative. Spheroids look too small. Authors does not provide detailed experimental setup of culture (time of culture, end point size of spheroids). Here is an example of how spheroids of MDA-MB-231 look like: https://www.frontiersin.org/articles/10.3389/fonc.2020.01543/full
New images demonstrating solid 3D structures need to be provided.
Figure 1B image is not high quality and is pixelated once enlarged, please provide better image for assessment.
Figure 2: Beside qPCR, authors should provide western blot analysis of complex 1 genes expression.
Fig 2A - authors should explain what conditions MCF10A were grown before the analysis. Difference in ROS levels between wild type and RORalfa knockdown does not seem to be significant, as gating approach seems to be arbitrary and designed to "fit the need", rather than demonstrates actual biological difference between two conditions.
Fig. 4B - Clearly indicate what is exactly presented in ChIP qPCR and present data as percentage of enrichment over the input and explain the calculation. There is no description of how Flag tagged version of ROR alfa cell line was generated. Please provide detailed description of what vector, cloning and expression strategy was used to generate these cells.
Fig 4C - Authors need to explain what datasets they are using and give proper references.
Fig 5C - Time zero of injection has to be provided to evaluate luciferase activity at the start of the experiment. As mice were imaged once per week, please provide images of consecutive weekly measurements. It can be provided as supplementary material.
Figure 7 - Transwell images in Fig7A and Fig7B are very faint and pixelated, authors should improve quality of images and present black and white images and increase contrast.
Author Response
1. Figure 1: A) Images and structure of 3D spheroids does not look representative. Spheroids look too small. Authors does not provide detailed experimental setup of culture (time of culture, end point size of spheroids). Here is an example of how spheroids of MDA-MB-231 look like: https://www.frontiersin.org/articles/10.3389/fonc.2020.01543/full
Response: Thank the reviewer for this and following comments. We have provided information related time of culture and size (scale bar) in the figure legend for these images. We followed the 3D culture protocol developed by Dr. Mina J Bissell’s lab. It has been shown that MDA-MB-231 cells form stellate structure in 3D Matrigel (Mol Oncol. 2007 Jun;1(1):84-96.). Our results are consistent with the previous publications. The 3D culture that reviewer pointed out was in 3D agarose gel, which is different from our assay.
2. Figure 1B image is not high quality and is pixelated once enlarged, please provide better image for assessment.
Response: We have uploaded new images in figure 1B and listed the gene names in supplementary Table 1.
3. Figure 2: Beside qPCR, authors should provide western blot analysis of complex 1 genes expression.
Response: We have performed the western blot analysis to examine NDUFA11 protein levels in control and RORα-expressing MDA-MB-231 cells. The data are now included in Supplemental figure 3.
4. Fig 2A - authors should explain what conditions MCF10A were grown before the analysis. Difference in ROS levels between wild type and RORalfa knockdown does not seem to be significant, as gating approach seems to be arbitrary and designed to "fit the need", rather than demonstrates actual biological difference between two conditions.
Response: MCF-10A cells were cultured in DMEM/F12 with 5% horse serum, 20 ng/ml EGF, 0.5 mg/ml hydrocortisone, 100 ng/ml cholera toxin, 10 μg/ml insulin and 1% Pen/Strep. This information has been included in the material and methods. The top 50% of cell population in control group was gated as “ROS High”. The x-axis was labeled in log scale, and the difference of two groups was statistically significant.
5. Fig. 4B - Clearly indicate what is exactly presented in ChIP qPCR and present data as percentage of enrichment over the input and explain the calculation. There is no description of how Flag tagged version of ROR alfa cell line was generated. Please provide detailed description of what vector, cloning and expression strategy was used to generate these cells.
Response: Flag-tagged RORA cDNA were cloned into pCDH1 plasmid and generated expression vector pCDH1-RORα-Flag. HEK293 FT cells were transfected with pCDH1 or pCDH1-RORα-Flag plus packaging lentivector using FuGENE (Promega, 0000356676). MDA-MB-231 cells were infected with lentivirus and selected by puromycin 48h after infection. This information has been included in the material and methods.
6. Fig 4C - Authors need to explain what datasets they are using and give proper references.
Response: The gene expression data were derived from TCGA dataset. This information is now included in the results.
7. Fig 5C - Time zero of injection has to be provided to evaluate luciferase activity at the start of the experiment. As mice were imaged once per week, please provide images of consecutive weekly measurements. It can be provided as supplementary material.
Response: We have injected same amount of control and RORa-expressing cancer cells. We are sorry that we did not take time zero images. We have provided weekly images in Supplemental figure 3.
8. Figure 7 - Transwell images in Fig7A and Fig7B are very faint and pixelated, authors should improve quality of images and present black and white images and increase contrast.
Response: We have uploaded new images for Fig7A and Fig7B.
Reviewer 4 Report
The objective of this study was to investigate the significance of orphan nuclear receptor alpha (RORα) in breast cancer. The manuscript " RORα suppresses cancer-associated inflammation by repressing respiratory complex I-dependent ROS generation" discussed RORα as a potential target to suppress ROS-induced tumor progression. The authors showed that RORα decreases ROS levels and inhibits ROS-mediated cytokine expression in breast cancer cells. The role of RORα in the suppression of mitochondrial complex 1 genes at transcription level was suggested as a potential mechanism behind RORα mediated inhibition of reactive oxygen species (ROS) generation. Also, the inhibitory role of RORα in mammary tumor metastasis and macrophage infiltration was reported. These findings are interesting and the manuscript is well-written and, well-structured. I believe that the study's objectives are interesting and fit well within the scope of the journal and that the experiment was carried out with an appropriate description of the methods utilized. Thus, in my opinion, the manuscript merits the publication in International Journal of Molecular Sciences.
However, the authors should address the following issues
Comments:
- In the abstract section, it would be appropriate to mention why the authors investigated the RORα
- Authors need to carefully check the material and method section. Include catalog numbers wherever appropriate.
- Figure 1 B is very difficult to read since the font size is small. As it represents one of the most essential data, the authors must adjust the figure for readability.
- Figure 1D shows the labels for the targets, but the names of the samples are absent.
- Is it appropriate to include figure legends under respective supplementary figures?
- The authors wrote “Bar: 100 µm” in figure legend 1 (A). The bars, however, are not visible/added in all sub-pictures.
- Is it necessary to culture the HMT-3522 S1 cells without antibiotics?
- line 472, correct repeated word “female”.
Author Response
1. In the abstract section, it would be appropriate to mention why the authors investigated the RORα
Response: Thank the reviewer for this comment. We now include one sentence in the abstract to explain why we investigated RORa.
2. Authors need to carefully check the material and method section. Include catalog numbers wherever appropriate.
Response: We have checked this section and included catalog number for recombinant human IL-6, NAC, DMEM, DMEM/F12, FBS, Pen/Strep, Matrigel, Non-Essential Amino Acids, L-glutamine and Sodium Pyruvate.
3. Figure 1 B is very difficult to read since the font size is small. As it represents one of the most essential data, the authors must adjust the figure for readability.
Response: We uploaded new images and listed the gene names in supplementary Table 1.
4. Figure 1D shows the labels for the targets, but the names of the samples are absent.
Response: We have included the names of the samples.
5. Is it appropriate to include figure legends under respective supplementary figures?
Response: We have included figure legends under supplementary figures.
6. The authors wrote “Bar: 100 µm” in figure legend 1 (A). The bars, however, are not visible/added in all sub-pictures.
Response: We have uploaded new images with bar in figure 1A.
7. Is it necessary to culture the HMT-3522 S1 cells without antibiotics?
Response: This cell line has been cultured without antibiotics since it was developed at Dr. Mina J. Bissell’s lab. We followed the protocol from Bissell’s lab and keep culturing them without antibiotics.
8. line 472, correct repeated word “female”.
Response: We have deleted the repeated word “female”.
Round 2
Reviewer 3 Report
Thank you for the update. Some revisions does improve the manuscript, however some parts remain shaky.
Figure 3 A and B. It is still unclear to me and authors does not address this in their response - how ChIP results were calculated. What was percentage of the Input that was used in ChIP qPCR runs? What is data normalized to? If data is presented as a fold change, what is fold over? Which regions of promoter were used for ChIP? This all has to be reflected in figure and figure legend.
Figure 5C - the only way for the reviewers and readers to critically evaluate whether comparable amount of cells and luciferase signal was present at the start of the experiment is to see time zero. Authors provide images only after 2 weeks past injection, what happened with 1 week image? Also, how can they explain fluctuating number of mice in the images - there are 2 mice in control group after 2 weeks, just for the third mouse to reappear in week 3 and disappear again at week 4. Mouse in ROR overexpressing group disappears at week 3 and comes back again at week 6 and 7? What is an explanation for this?
Author Response
Figure 3 A and B. It is still unclear to me and authors does not address this in their response - how ChIP results were calculated. What was percentage of the Input that was used in ChIP qPCR runs? What is data normalized to? If data is presented as a fold change, what is fold over? Which regions of promoter were used for ChIP? This all has to be reflected in figure and figure legend.
Response: We apologize that we did not address this comment in the previous response. The ChIP assay was performed based on the Upstate Biotechnology ChIP protocol, with a few modifications as shown in our previous study (JBC, 2007). 10% of extracted chromatin was aliquoted as input, and 5% of input was used as template for qPCR. After normalization to the input, we calculated the relative enrichment of ChIP DNA in “RORα-Flag” group to the control group. This information has been included in the Figure legend and the material/method.
We now include the detail information of RORE sites and primers in NDUFA11 and NDUFS6 promoters in Fig. 4A. Red boxes are potential RORE in their promoter regions. Primers for qPCR were designed based on the sequence before and after ROREs. The sequences of these primers are included in material and methods.
Figure 5C - the only way for the reviewers and readers to critically evaluate whether comparable amount of cells and luciferase signal was present at the start of the experiment is to see time zero. Authors provide images only after 2 weeks past injection, what happened with 1 week image? Also, how can they explain fluctuating number of mice in the images - there are 2 mice in control group after 2 weeks, just for the third mouse to reappear in week 3 and disappear again at week 4. Mouse in ROR overexpressing group disappears at week 3 and comes back again at week 6 and 7? What is an explanation for this?
Response: We agree with the reviewer that luciferase signal at day zero would provide important information about RORa function in regulating the initiation of colonization. Unfortunately, we encountered a technique issue to acquire these data. We often see that mice are weak after tail vein injection, and we usually put them on the warm pad right after injection for 24 hours to let them recover. Therefore, we did not inject luciferin in the mice and perform the luciferase measurement at time zero. We started to monitor colonization of cancer cells from week two. According to our experience, no specific luciferase signals can be detected before week three.
As for the luciferase images in the supplemental figures, we had six mice in each group. Either two or three mice were put in the device each time for taking luciferase images (the device can only hold three mice each time). However, the quantification data were from all six mice. This information has been included in the figure legend. We are sorry that we did not clearly explain this in our previous response.